# HOX13-dependent chromatin accessibility underlies the transition towards the digit development program

Ines Desanlis [1,2,9], Yacine Kherdjemil [1,2,7,9], Alexandre Mayran [3,4,9], Yasser Bouklouch[1], Claudia Gentile[1,5,8], Rushikesh Sheth[6], Rolf Zeller[6], Jacques Drouin[2,3] & Marie Kmita [1,2,5 ✉]

*Hox* genes encode transcription factors (TFs) that establish morphological diversity in the developing embryo. The similar DNA-binding motifs of the various HOX TFs contrast with the wide-range of HOX-dependent genetic programs. The influence of the chromatin context on HOX binding specificity remains elusive. Here, we used the developing limb as a model system to compare the binding specificity of HOXA13 and HOXD13 (HOX13 hereafter), which are required for digit formation, and HOXA11, involved in forearm/leg development. We find that upon ectopic expression in distal limb buds, HOXA11 binds sites normally HOX13-specific. Importantly, these sites are loci whose chromatin accessibility relies on HOX13. Moreover, we show that chromatin accessibility specific to the distal limb requires HOX13 function. Based on these results, we propose that HOX13 TFs pioneer the distal limb-specific chromatin accessibility landscape for the proper implementation of the distal limb developmental program.

[1] Genetics and Development Research Unit, Institut de Recherches Cliniques de Montréal, Montréal, Québec, Canada. [2] Département de Médecine, Université de Montréal, Montréal, QC, Canada. [3] Molecular Genetics Research Unit, Institut de Recherches Cliniques de Montréal, Montréal, Québec, Canada. [4] EPFL, School of Life Sciences, 1015 Lausanne, Switzerland. [5] Department of Experimental Medicine, McGill University, Montreal, Quebec, Canada. [6] Developmental Genetics, Department of Biomedicine, University of Basel, 4058 Basel, Switzerland. [7] Present address: EMBL Heidelberg, Meyerhofstrasse 1, D-69117 Heidelberg, Germany. [8] Present address: Dana-Farber Cancer Institute and Harvard Medical School, 450 Brookline Avenue, Boston, MA 02215, USA. [9] These authors contributed equally: Ines Desanlis, Yacine Kherdjemil, Alexandre Mayran. Authors are listed in alphabetical order. ✉email: Marie.Kmita@ircm.qc.ca

The differential expression of the *Hox* genes along the main axis of the developing embryo establishes positional information that instructs cells about their fate, ultimately generating morphological diversity[1–3]. Consequently, spatial or temporal deregulation of *Hox* genes can cause severe alterations of the body plan[3]. In most Vertebrates, *Hox* genes are grouped in four clusters, referred to as *HoxA–D* clusters. The coordinated regulation of the *Hox* genes is associated with their cluster organization, whereby the sequential activation of the *Hox* genes follows the gene order on the chromosome (groups 1–13). The wide-range of morphological features associated with the distinct HOX TFs contrasts with the similarity in their DNA-binding motifs[4]. Several studies uncovered the relevance of co-factors, notably the MEIS and PBX family of TFs, in increasing DNA-binding specificity[4–6]. Comparison of genome-wide binding of the HOXA and HOXD TFs relevant to limb development, assessed following retroviral infection of individual *Hox* gene in chicken mesenchymal limb bud cells, further substantiated the functional significance of co-factors in HOX-binding specificity[7]. Strikingly, principal component analysis of HOX DNA binding in this primary cell culture system also revealed that group 11 HOX TFs and HOXD13 cluster in the same subgroup[7] despite the fact that HOXA11/HOXD11 are required for the zeugopod (forearm) developmental program[8], while HOXA13 and HOXD13 are mandatory for digit development[9] (Fig. 1a). Such apparent discrepancies between the binding profiles in micromass culture and the function of group 11 and 13 HOX TFs in the developing limb may be related to cell-type dependent DNA-binding specificity, as HOXA11 and HOX13 are expressed in distinct domains of the limb bud. Interestingly, Porcelli et al.[10] by analyzing binding specificity of *Drosophila* HOX factors transfected in Kc167 cells, provided evidence that HOX factors vary in their capacity of binding chromatin in an inaccessible state. In turn, it suggests that cell type-specific differences in the chromatin accessibility landscape may change the genome-wide binding of HOX factors in a cell type-specific manner, at least for those HOX factors unable to bind inaccessible chromatin. More generally, it raises the question of the extent to which changes in the chromatin accessibility landscape modulate the target repertoire of the HOX TFs spatially/temporally during embryogenesis.

To address this question, we investigated the impact of cellular context on the genome-wide binding of HOXA11 and HOX13, which are normally expressed in mutually exclusive domains in the developing limb (Fig. 1a) but have highly similar homeodomain sequences (e.g., ref. [7]; Fig. 1b). Our data reveal that the HOX13-specific targets, as compared to the HOXA11-binding repertoire, are loci bound by HOXA11 when HOXA11 is expressed in the HOX13 limb domain. This tissue-specific variation of the HOXA11 binding repertoire is associated with changes in the chromatin accessibility landscape triggered by HOX13. Importantly, our data reveal that the HOX13-dependent gain in accessibility at HOX13 targets corresponds to the gain in accessibility that accompanies the transition from proximal to distal limb.

## Results

**HOXA11 binding varies in a tissue-specific manner.** To investigate the impact of the cellular context on the target repertoire of HOXA11 and HOX13, we first analyzed the genome-wide binding of HOXA11 in wild-type limb buds at embryonic day 11.5 (E11.5), by performing chromatin immunoprecipitation followed by high-throughput sequencing (ChIP-seq) and compared the data with those previously reported for HOX13 (ref. [11]) (Fig. 1c; Supplementary Fig. 2). Reminiscent of the distribution of HOX13-bound loci in distal limbs[11], the vast

majority of HOXA11 ChIP-seq peaks are located at genomic regions distinct from transcriptional start site (TSS; Supplementary Fig. 2h). While most peaks are common in both data sets, 23% of HOXA11 (i.e., 2128) or HOX13 (i.e., 2130) targets are located at distinct loci (Fig. 1c) and de novo motif search uncovered slight differences in HOX motifs for common vs. specific targets (Fig. 1d). We next tested whether the cellular environment could modulate the specificity distinguishing HOXA11 from HOX13-binding sites. To this aim, we analyzed HOXA11-binding profile in *Prrx1:Cre;Rosa26<sup>Hoxa11/Hoxa11</sup>* limb buds in which *Hoxa11* is ectopically expressed distally[12] (*R26<sup>A11d/A11d</sup>* in Fig. 1e). We observed that the loci identified as HOX13-specific targets in the wild-type limb bud are bound by HOXA11 in *R26<sup>A11d/A11d</sup>* limb buds (Fig. 1f). This result suggested that the primary driver for the distinct HOXA11- and HOX13-binding profiles in wild-type limb buds is associated with their expression in distinct cell populations. To further assess whether this ectopic HOXA11 binding is due to the presence of HOX13 or triggered by other factors specific to distal limb cells, we took advantage of the *Hox13<sup>−/−</sup>* mutant[9] in which *Hoxa11* is distally expressed[11,13,14]. Analysis of HOXA11 genome-wide binding in *Hox13<sup>−/−</sup>* limbs revealed that the ectopic HOXA11 binding at HOX13 targets observed in *R26<sup>A11d/A11d</sup>* limbs were absent in *Hox13<sup>−/−</sup>* limbs (Fig. 1f, Supplementary Fig. 3a) while the wild-type HOXA11-binding pattern was unaffected (Supplementary Fig. 3b, c). This result indicated that the capacity of HOXA11 to bind new loci when expressed in the distal limb bud is dependent on HOX13 function.

**HOX13 pioneer action expands HOXA11-binding repertoire.** To assess whether chromatin accessibility may influence HOXA11 and HOX13 binding, we performed assay for transposase accessible chromatin followed by high-throughput sequencing (ATAC-seq[15]) on proximal and distal wild-type limb buds. We identified 1629 sites showing stronger accessibility in distal limb buds (Fig. 2a, Supplementary Fig. 4a). The prevalent motif associated with these sites is the previously identified HOX13 motif (Fig. 2b), while distinct motifs are found at proximal limb-enriched ATAC peaks (Supplementary Fig. 4b). Accordingly, HOX13 TFs bind distal limb-enriched accessible sites and not the proximal limb-specific accessible sites (Fig. 2c). Moreover, these distal limb sites tend to be associated with genes involved in digit morphogenesis, supporting their involvement in the distal limb program (Supplementary Fig. 4c–f). Previous analysis of HOX13-dependent gene regulation revealed that the transition from the early to the distal/late limb developmental program relies on HOX13 function and it was proposed that this switch in developmental program could be associated with a potential pioneer activity of the HOX13 TFs[11]. The finding that cellular environment modulates HOXA11 binding in a HOX13-dependent manner (Fig. 1) prompted us to test whether the ectopic HOXA11 binding at loci normally specifically bound by HOX13 could originate from changes in chromatin accessibility mediated by HOX13 TFs. We thus performed ATAC-seq on wild type and *Hox13<sup>−/−</sup>* distal limbs (Supplementary Fig. 5a) and identified substantial changes in chromatin accessibility (Fig. 2d). Interestingly, loci with decreased accessibility in *Hox13<sup>−/−</sup>* distal limbs were found primarily associated with a motif highly similar to the HOX13 motif (Fig. 2e) and were loci with distal limb-specific chromatin accessibility in the wild-type context, which for the vast majority were bound by HOX13 (Fig. 2f). In contrast, loci with increased accessibility in *Hox13<sup>−/−</sup>* distal limbs were poorly correlated with the HOX13 ChIP-seq peaks identified in the wild-type context, suggesting that this gain in chromatin accessibility is an indirect outcome of *Hox13* inactivation (Supplementary

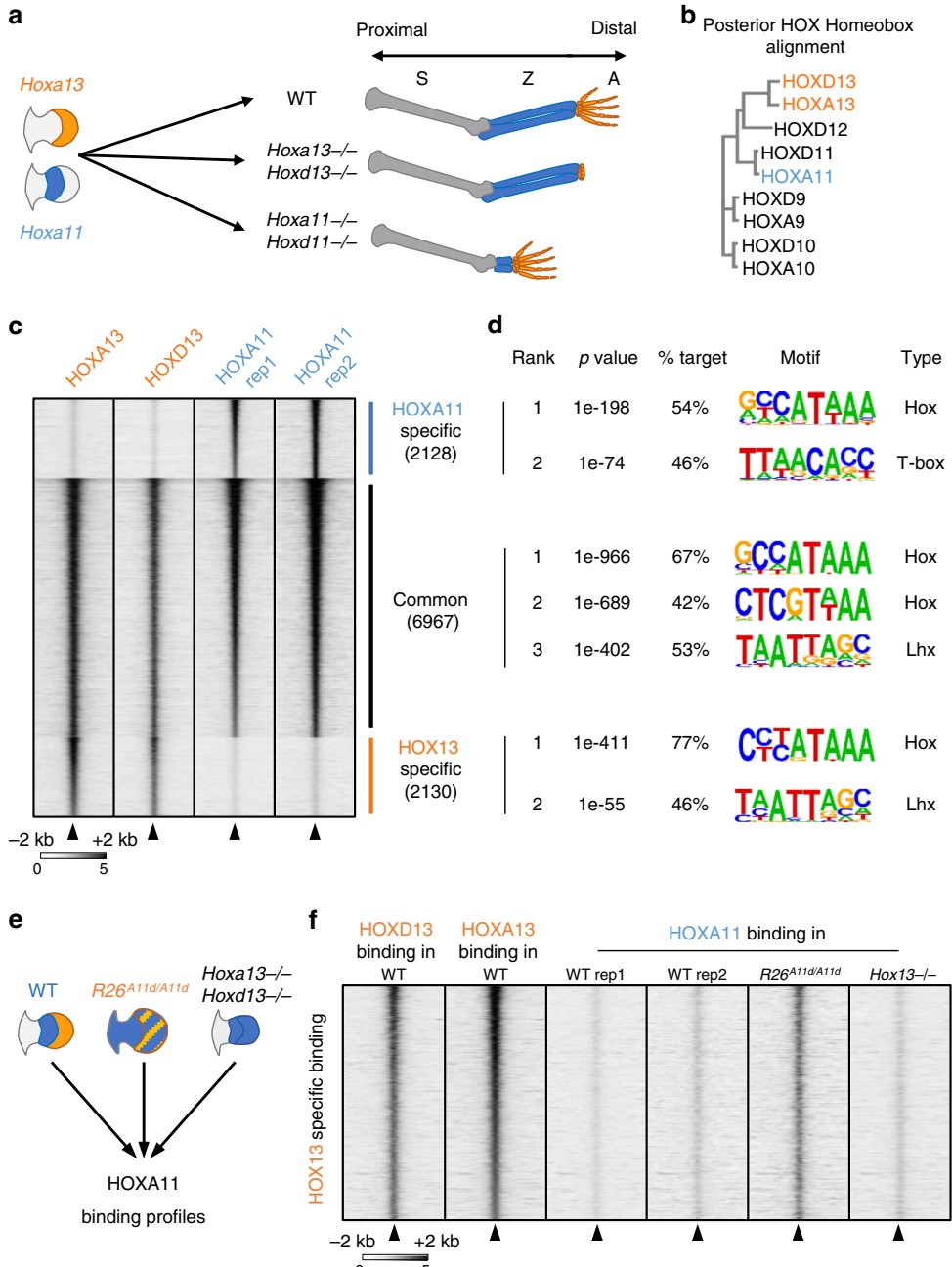

**Fig. 1 Loss of HOX13 specificity upon *Hoxa11* expression in distal limb buds. a** Schematics illustrating the mutually exclusive expression pattern of *Hoxa11* and *Hoxa13* in mouse limb buds (left) as well as the forelimb phenotype associated with the inactivation of *Hoxa/d11* and *Hoxa/d13* (right). **b** Alignment of 5′ *Hoxa* and *Hoxd* homeobox sequences revealing the close relationship between HOXA/D11 and HOXA/D13. **c** Heatmap showing ChIP-seq read density of HOXA11 (blue) and HOXA/D13 (orange) at all peaks for HOXA11 and HOXA13 with a *p* value < $10^{-20}$, in a 4 kb window. Peaks are ranked based on *p* value (Poisson distribution *p* value based on lambda and corrected for multiple comparison using the Benjamini–Hochberg correction computed by MACS2) and according to the binding specificity. Color scale indicates reads per million reads (RPM). **d** Top scoring motif uncovered using HOMER de novo motif analysis in a 200 bp window around the peaks center, for the three categories depicted in panel (**c**). **e** Schematics showing the expression pattern of *Hoxa11* (blue) and *Hoxa13* (orange) in wild type, *Prrx1Cre;Rosa*$^{Hoxa11/Hoxa11}$ mouse (*R26*$^{A11d/A11d}$) and *Hoxa13*$^{-/-}$;*Hoxd13*$^{-/-}$ (*Hox13*$^{-/-}$) mouse limb buds. These were used to assess binding of HOXA11. **f** Heatmaps showing a 4 kb window of ChIP-seq read density for HOXA13 and HOXD13 in wild type (left), HOXA11 in wild type (middle), *R26*$^{A11d/A11d}$ and *Hox13*$^{-/-}$ (right) E11.5 forelimb buds at HOXA13-specific peaks with a *p* value < $10^{-20}$. Peaks are ranked based on *p* value of HOXA13 binding (Poisson distribution *p* value based on lambda and corrected for multiple comparison using the Benjamini–Hochberg correction computed by MACS2). Color scale indicates reads per million reads (RPM). Source data are provided as a Source Data file.

Fig. 5b, c). Together, these data provide evidence that the gain of chromatin accessibility during the transition from proximal to distal limb is directly associated with HOX13 function. Moreover, we found that in *R26*$^{A11d/A11d}$ limb buds, the ectopic HOXA11 binding at sites which are HOX13-specific in wild-type limb buds,

are loci for which chromatin accessibility relies on HOX13 (Fig. 2g–i). Importantly, this ectopic HOXA11 binding was lost in *Hox13*$^{-/-}$ limb buds (Fig. 1f), even though *Hoxa11* is expressed in distal cells normally expressing *Hox13*[13]. Together, our results suggested that, in *R26*$^{A11d/A11d}$ limb buds, HOX13-dependent

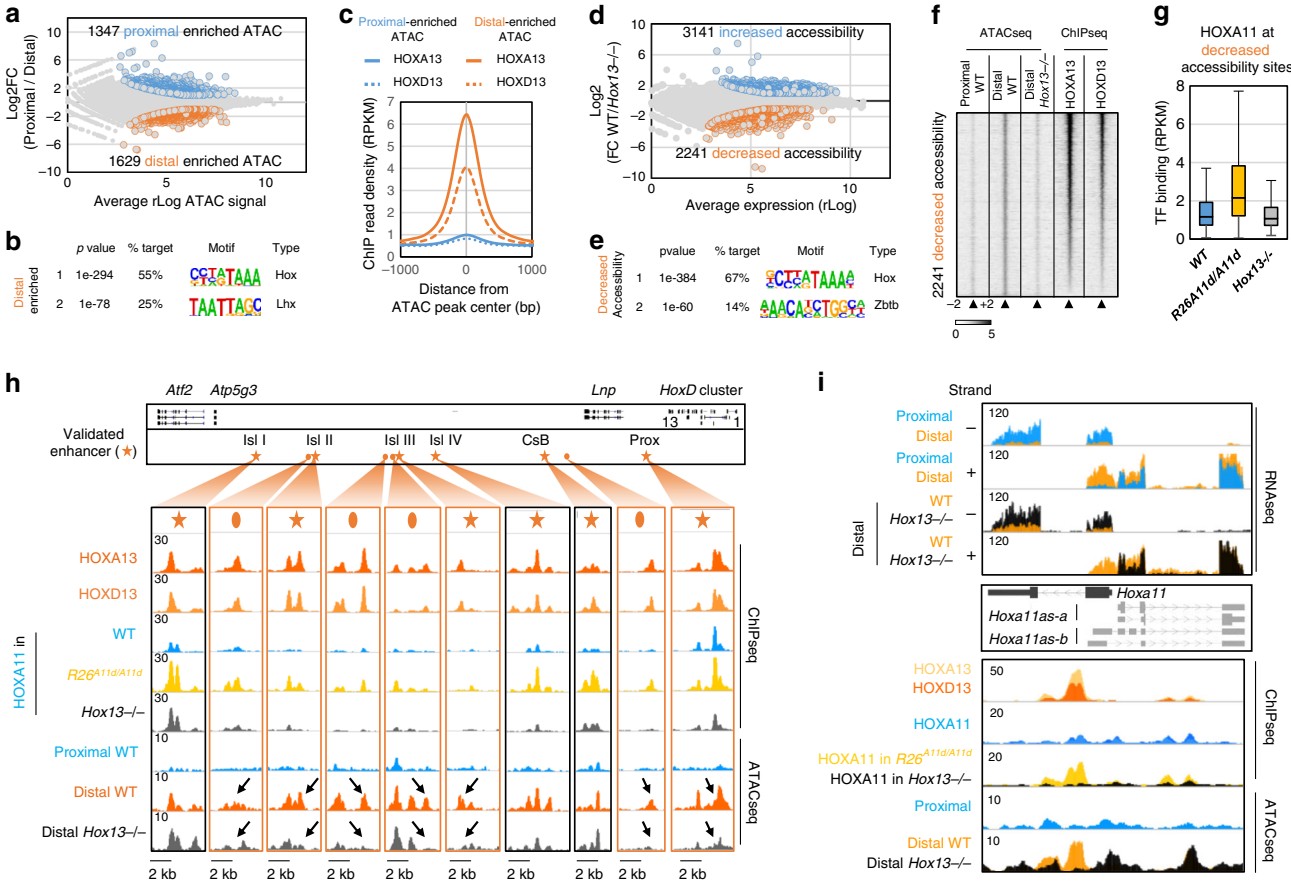

**Fig. 2 HOX13-specific binding in wild-type limbs correlates with HOX13-dependent chromatin accessibility. a** MA Plot showing the average rLog (computed using DESeq2) of ATAC signal (x-axis) over the log2 fold change of proximal versus distal limb buds ATAC-seq peaks. Proximal- (blue) and distal-enriched accessibility (orange) are selected by the adjusted p value < 0.05 (computed by DESeq2 using Benjamini–Hochberg algorithm) and fold change > ±2. **b** Top scoring motifs found by HOMER de novo motif analysis on the whole peak window at the distally enriched peaks. **c** Average profile showing HOXA/D13 ChIP-seq signal (RPKM) at proximal (blue) and distal (orange) enriched ATAC peaks. **d** MA Plot showing the average rLog (computed using DESeq2) of ATAC signal (x-axis) over the log2 fold change of wild type vs. $Hox13^{-/-}$ distal limb buds ATAC-seq peaks. Wild type (blue) and $Hox13^{-/-}$ (orange) enriched accessibility are selected by the adjusted p value < 0.05 (computed by DESeq2 using Benjamini–Hochberg algorithm) and fold change > ±2. **e** Top scoring motifs found by HOMER de novo motif analysis on the decreased accessibility peaks. **f** Heatmaps showing a 4 kb window of ATAC-seq read density for proximal (left) and distal in wild type and $Hox13^{-/-}$ (middle), and of ChIP-seq read density for HOXA13 and HOXD13 in wild-type (right) limb buds at decreased accessibility specific peaks with a p value < $10^{-20}$. Peaks are ranked based on p value of HOXA13 binding (Poisson distribution p value based on lambda and corrected for multiple comparison using the Benjamini–Hochberg correction computed by MACS2). Color scale indicates reads per million reads (RPM). **g** Boxplot showing the ChIP-seq signal read density (RPKM) for HOXA11 in wild type, in $R26^{A11d/A11d}$ and in $Hox13^{-/-}$ limb buds over a 200 bp window at sites with decreased accessibility. Center lines show medians; box limits indicate the 25th and 75th percentiles; whiskers extend to 1.5 times the interquartile range from the 25th to 75th percentiles. **h** Representation of the genomic region encompassing previously validated enhancers (orange stars) responsible for HoxD distal limb expression (upper panel). Genome browser views (IGV, bottom) of HOXA13, HOXD13, and HOXA11 ChIP-seq as well as ATAC-seq signals in wild-type proximal and distal and in $Hox13^{-/-}$ distal limb buds. Loci with reduced chromatin accessibility upon Hox13 inactivation are orange-framed and indicated by the arrow. Note loci with HOX13 ChIP-seq peaks, showing distal-specific, HOX13-dependent chromatin accessibility (orange ovals). **i** Genome browser views (IGV) of overlaid strand specific RNA-seq at the Hoxa11 locus in the indicated tissues (upper panel). Genome browser view (IGV) of the ChIP-seq and ATAC-seq data at the Hoxa11 locus in wild type, $R26^{A11d/A11d}$ and in $Hox13^{-/-}$ limb buds. Source data are provided as a Source Data file.

chromatin accessibility allows HOXA11 binding at loci that are HOX13-specific in the wild-type limb bud, i.e., in a context where Hox13 and Hoxa11 are expressed in mutually exclusive domains. These loci included several previously characterized distal limb enhancers. For instance, several enhancers controlling HoxD gene expression in distal limb[16–18] were identified as loci with ectopic HOXA11 binding in $R26^{A11d/A11d}$ limb buds and with decreased accessibility in absence of HOX13 function (Fig. 2h). This HOX13-dependent chromatin accessibility is also consistent with the loss of transcriptional activity previously reported at these enhancers in $Hox13^{-/-}$ limbs[11,19]. Similarly, chromatin accessibility and HOXA11 binding dependent on HOX13 were observed

at the enhancer driving Hoxa11 antisense transcription in distal limb (Fig. 2i), whose function is associated with digit patterning[12].

**Distal limb-specific chromatin accessibility requires HOX13.**
Next, we examined HOX13-dependent chromatin accessibility at the single-cell level to assess whether HOX13 pioneering action shows cell type specificity. To this aim, we performed single-cell ATAC-seq (scATAC-seq) using wild type and $Hox13^{-/-}$ E11.5 forelimb buds (Supplementary Fig. 6). Cell clustering was based on the similarity of their genome-wide chromatin accessibility and the resulting clusters (Fig. 3a) were annotated using

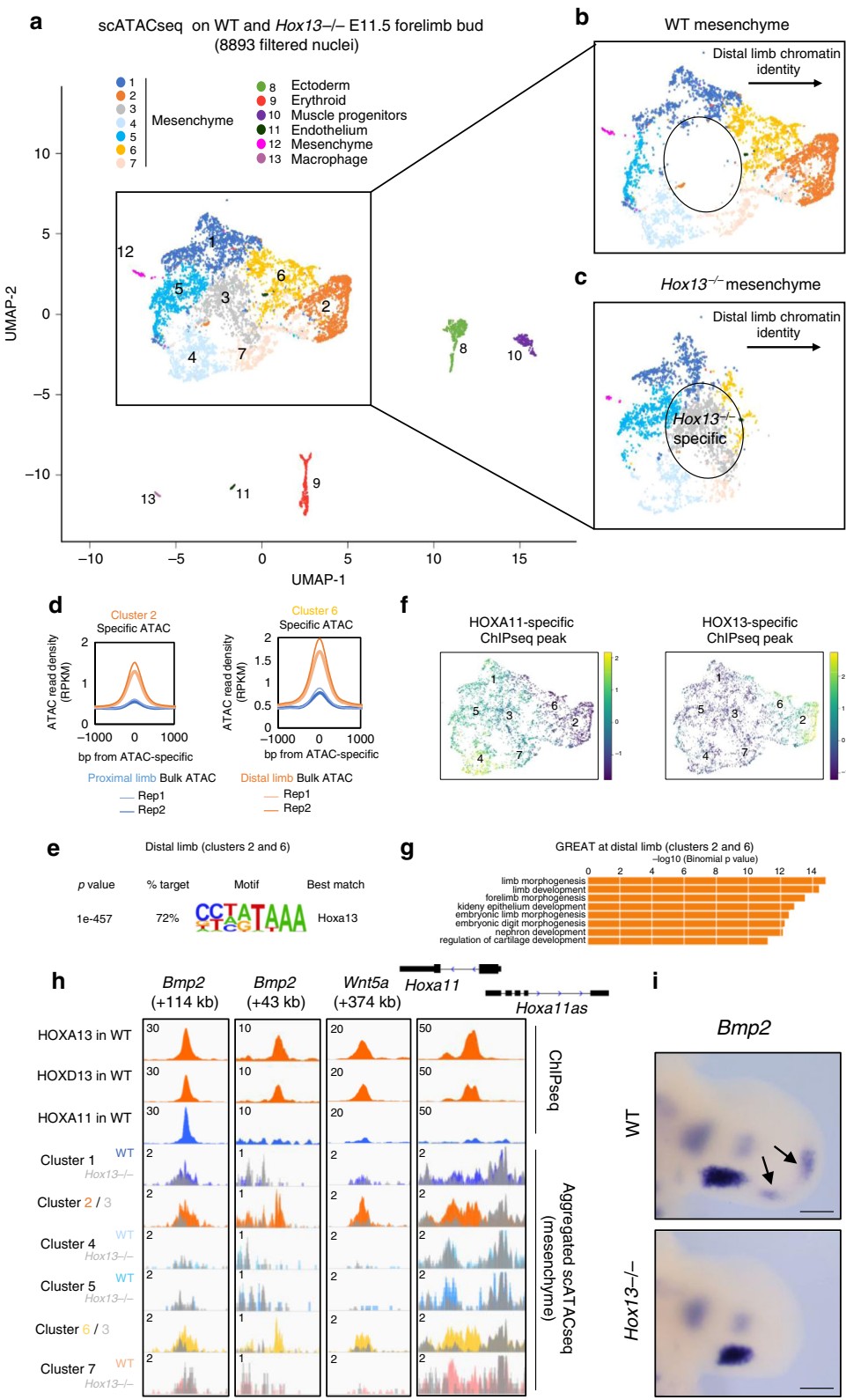

**a** scATACseq on WT and *Hox13*–/– E11.5 forelimb bud (8893 filtered nuclei)

Mesenchyme: 1, 2, 3, 4, 5, 6, 7

8 Ectoderm
9 Erythroid
10 Muscle progenitors
11 Endothelium
12 Mesenchyme
13 Macrophage

**b** WT mesenchyme — Distal limb chromatin identity

**c** *Hox13*–/– mesenchyme — Distal limb chromatin identity — *Hox13*–/– specific

**d** Cluster 2 Specific ATAC / Cluster 6 Specific ATAC. ATAC read density (RPKM), bp from ATAC-specific. Proximal limb Bulk ATAC (Rep1, Rep2); Distal limb Bulk ATAC (Rep1, Rep2)

**f** HOXA11-specific ChIPseq peak / HOX13-specific ChIPseq peak

**e** Distal limb (clusters 2 and 6)

| p value | % target | Motif | Best match |
|---|---|---|---|
| 1e-457 | 72% | CCTATAAA | Hoxa13 |

**g** GREAT at distal limb (clusters 2 and 6). –log10 (Binomial p value): limb morphogenesis, limb development, forelimb morphogenesis, kidney epithelium development, embryonic limb morphogenesis, embryonic digit morphogenesis, nephron development, regulation of cartilage development

**h** *Bmp2* (+114 kb), *Bmp2* (+43 kb), *Wnt5a* (+374 kb), *Hoxa11* / *Hoxa11as*

HOXA13 in WT, HOXD13 in WT, HOXA11 in WT (ChIPseq); Cluster 1 (WT / *Hox13*–/–), Cluster 2/3, Cluster 4 (WT / *Hox13*–/–), Cluster 5 (WT / *Hox13*–/–), Cluster 6/3, Cluster 7 (WT / *Hox13*–/–) (Aggregated scATACseq (mesenchyme))

**i** *Bmp2* — WT / *Hox13*–/–

aggregated accessibility profiles of several marker genes (Supplementary Fig. 7). Consistent with *Hox13* being expressed in the limb mesenchyme, we observed changes in the clustering of *Hox13*–/– cells only within the mesenchymal cell population (Fig. 3b, c, Supplementary Fig. 8a, b). Among the wild-type mesenchymal cell clusters, clusters 2 and 6 were the only clusters with an average profile specific for the distal limb, as identified

from the bulk ATAC-seq from distal limb buds (Fig. 3d). These two clusters were absent in the scATAC-seq data set from *Hox13*–/– limb (Fig. 3c), further supporting that HOX13 function is required to implement the distal limb-specific chromatin accessibility profile. De novo motif analysis at cluster-specific scATAC-seq peaks uncovered the HOXA13 motif for the clusters 2 and 6 (Fig. 3e) while distinct motifs were found for the other

**Fig. 3 Single-cell ATAC-seq analysis reveals specific clusters with modified chromatin accessibility upon *Hox13* inactivation. a** Uniform manifold approximation and projection (UMAP) representation of the distribution of the various clusters identified using single-cell ATAC-seq (scATAC-seq). Chromatin accessibility at promoter of genes known to be exclusively expressed in a specific cell population was used to assign cell identity to the clusters. **b, c** Magnification of mesenchymal clusters in wild type (top) and *Hox13*[−/−] (bottom). Cluster 2 (orange) is entirely lost in the *Hox13*[−/−] limb bud and cluster 6 (yellow) is drastically reduced, while cluster 3 (gray) is specific to *Hox13*[−/−] limb buds. Note that clusters corresponding to non-mesenchymal cells (8–13) are unaffected by *Hox13* inactivation. **d** Average profiles showing cluster 2 (left) and cluster 6 (right) ATAC read densities (RPKM) at proximal (blue) and distal (orange) enriched ATAC peaks. **e** Top scoring motif found by HOMER de novo motif analysis on the entire peak window at the distal limb (clusters 2 and 6) peaks. **f** HOXA11 (left) and HOX13 (right) specific ChIP-seq peaks represented by degree of overlap onto UMAP of mesenchyme cells from scATAC-seq of WT and *Hox13*[−/−] E11.5 limb buds (reproduced from panel (**a**)). **g** GREAT analysis showing the enriched GO terms (*p* value computed using binomial test over genomic regions by GREAT) for biological processes for genes associated with distal limb (clusters 2 and 6) scATAC-seq peaks. **h** Genome browser view (IGV) of a subset of previously identified distal limb enhancers or potential distal limb enhancers associated with genes expressed in distal limb buds. ATAC-seq peaks in cluster 2 (orange) and 3 (gray) are superimposed to highlight the loss of accessibility upon *Hox13* inactivation. **i** Whole-mount in situ hybridization of *Bmp2* in wild type and *Hox13*[−/−] E11.5 limb buds. The arrows point to the two spots of distal *Bmp2* expression lost in *Hox13*[−/−]. A minimum of three embryos per genotype was assayed for reproducibility (*n* = 3). The scale bar represents 200 µm. Source data are provided as a Source Data file.

mesenchymal clusters (Supplementary Fig. 9). Accordingly, HOX13-binding profile was highly similar to the chromatin accessibility profile of clusters 2 and 6 (Fig. 3f). Moreover, GREAT analysis indicated that only the accessibility profile of clusters 2 and 6 tends to be associated with digit morphogenesis (Fig. 3g; Supplementary Fig. 9). For instance, the enhancer driving *Bmp2* in distal limb[20] was in an open state in clusters 2 and 6 and lost its accessible state upon *Hox13* inactivation (cluster 3; Fig. 3h). Accordingly, *Bmp2* expression is lost in *Hox13*[−/−] distal limb while its proximal expression remained unaffected (Fig. 3i). Interestingly, the occurrence of a *Hox13*[−/−] specific cluster (cluster 3, Fig. 3c) suggested that chromatin accessibility did not simply switch to a proximal limb bud profile, consistent with the absence of distal to proximal transformation of *Hox13*[−/−] limb skeleton[9]. Together these results provided evidence that the process by which a subset of mesenchymal cells acquire the distal limb-specific chromatin accessibility profile directly relies on HOX13 function.

## Discussion

In this study, we use the developing limb as a model system to investigate the impact of cellular context on HOX-binding specificity. By comparing the genome-wide binding of HOX13 and HOXA11, we found that in the wild-type context about 23% of HOX13 targets are specific to HOX13 consistent with HOX13 being required for the implementation of the digit development program while HOXA11 contributes to the formation of the zeugopod. Our results provide evidence that this specificity relies on HOX13 and HOXA11 being expressed in mutually exclusive domains as the expression of HOXA11 in the HOX13 domain results in HOXA11 binding to the HOX13-specific targets. Importantly, we uncovered that this ectopic HOXA11 binding occurs only in a context where the HOX13-specific targets are in a chromatin accessible state and this open conformation requires HOX13 function. This capacity of HOX13 to trigger the switch from a closed chromatin conformation to an accessible one and to allow the binding of other TFs (Fig. 4), two defining properties of pioneer factors[21,22], raises the possibility that HOX13-dependent chromatin accessibility is key to the implementation of the HOX13-specific developmental program. Accordingly, the gain in chromatin accessibility observed in the transition from proximal to distal limb bud occurs at HOX13-specific targets and is dependent on HOX13 function (Fig. 4). Interestingly, binding of HOX13 to initially inaccessible chromatin was observed at the enhancer referred to as Prox, which drives *Hoxd* expression in the genital bud[23] (referred to as Prox in ref. [23]), raising the possibility that HOX13 pioneering action may not be limited to

the developing limb but rather a general feature of HOX13-dependent developmental programs.

Whether other HOX factors implement their specific developmental program by changing the chromatin accessibility landscape remains to be established. Interestingly, transfection of *Drosophila melanogaster* HOX factors into Kc167 cells shows that, in this system, a subset of HOX factors (e.g., Antp and Ubx) bind almost exclusively to targets in an accessible chromatin state while other (e.g., Abd-B) bind to loci initially in a closed conformation[10]. Similarly, a recent study provided evidence that, following transfection of HOXC factors in an in vitro motor neuron differentiation system, the divergence of their genome-wide binding profile is associated with differences in their ability to bind inaccessible chromatin[24]. These evidence that several HOX factors bind inaccessible chromatin together with the finding that, in vivo, HOX13 TFs establish the distal limb-specific chromatin accessibility landscape, suggest that the differential chromatin accessibility contingent on HOX factors is critical for implementing the distinct HOX-dependent developmental programs.

Our comparative study of HOXA11 and HOX13 in both wild-type and mutant limb buds shows that the HOX13-specific targets can be bound by HOXA11 when HOXA11 is expressed in the HOX13 domain, but not in absence of HOX13, when these targets are in a closed conformation. This might explain why expressing HOXA11 in the HOX13 domain results in the formation of additional digits while distal expression of HOXA11 in the *Hox13*[−/−] limb bud is unable to trigger digit development (Fig. 4). Our data also indicate that chromatin opening triggered by the HOX13 factors can expand the target repertoire of other HOX factors expressed in the same limb bud domain, which possibly contributes to the severity of the phenotypes associated with ectopic *Hox13* expression. For instance, the severe zeugopod malformation observed in the *Ulnaless/+* mice[25,26], a mutation corresponding to a genomic inversion resulting in *Hoxd13* ectopic expression in the presumptive zeugopod[27], may be associated with HOXD13 eliciting chromatin opening at enhancer elements, the transcriptional activity of which could be strengthen/modulated by the other HOX factors expressed in these cells.

The evidence that HOX13 function is required for chromatin accessibility at a subset of distal limb *HoxD* enhancers suggests that it contributes to the distal expansion of the *HoxD* expression as limb development proceeds. Interestingly, it was previously proposed that the increase/expansion of the *Hoxd13* expression domain might have been a driver for the origin of the novel endoskeletal elements characterizing the fin-to-limb transition[28]. In this view, the HOX13-dependent chromatin accessibility identified here for a subset of the *HoxD* enhancers might have

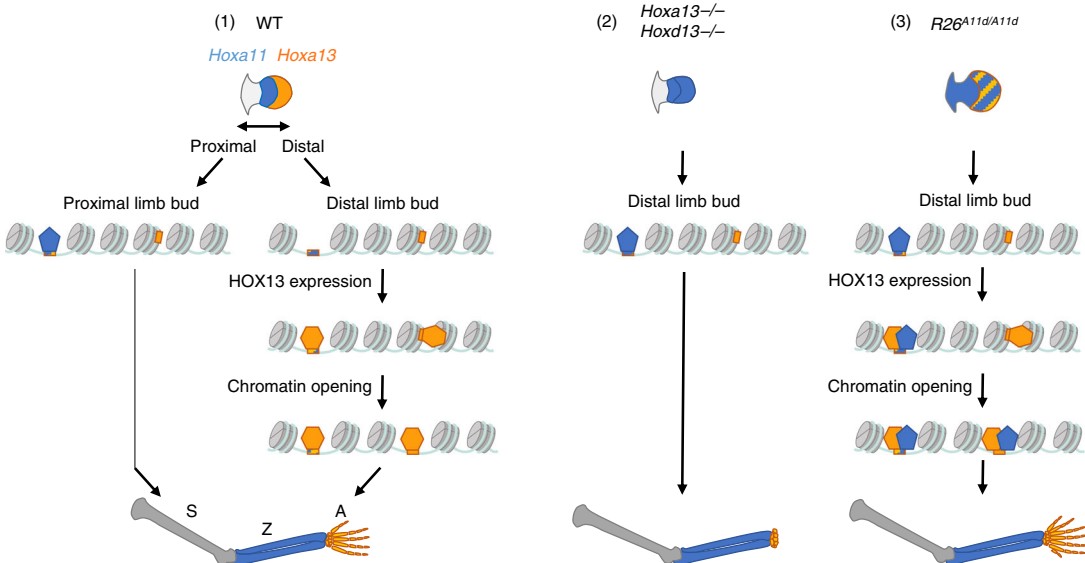

**Fig. 4 Model of the role of HOX13 in chromatin accessibility in the transition from proximal to distal limb.** In wild-type limb buds (left), HOXA11 (blue pentagons) is expressed in the proximal domain, whereas HOX13 factors (orange hexagons) are expressed in the distal domain. HOX13 TFs bind target loci irrespective of their chromatin accessibility, such that targets in a "closed" conformation switch to an accessible state. This change in chromatin accessibility does not occur in *Hox13* null limb buds (middle), resulting in the loss of the distal-specific chromatin accessibility landscape and disrupting the distal limb developmental program. In the *R26^{A11d/A11d}* mutant limb buds (right), HOXA11 is expressed in the distal limb domain (HOX13 domain) where its target repertoire expands to loci made accessible by HOX13. Such ectopic binding of HOXA11 at sites that are HOX13-specific in the wild-type context correlates with the formation of additional digits in the *R26^{A11d/A11d}* limbs. In absence of HOX13, HOXA11 is ectopically expressed in the distal bud but is unable to bind the distal-specific sites in a non-accessible chromatin state (middle).

been important to trigger the activity of evolutionary novel enhancers in the distal limb bud, which in turn have contributed to the emergence of digits. Similarly, the HOX13-dependent chromatin accessibility at the evolutionary novel enhancer driving *Hoxa11* antisense transcription, required to prevent distal *Hoxa11* expression and polydactyly[12,29], ensured a distal-specific modification of *Hoxa11* expression in tetrapods. Consistent with this view, we found that the enhancer driving *Hoxa11* antisense transcription is in an accessible state in the presumptive digit domain of chicken limb buds while it is in a closed configuration in early limb buds, when *Hox13* genes are not expressed (Supplementary Fig. 10). This result corroborates the findings in wild-type and mutant mouse limb buds and points to a conserved role of the HOX13 TFs in regulating chromatin accessibility during both mouse and chicken embryonic development. Based on these data, we propose that HOX13 pioneer activity added spatial constraints to the activity of evolutionary novel enhancer elements, which may have contributed to the broader morphological diversity of the distal limb as compared to the proximal limb.

## Methods

**Mouse lines**. The generation of *Hoxa13null* (*Hoxa13^{Str}*), *Hoxd13null* (*Hoxd13^{lacZ}*), and *Rosa^{Hoxa11}* mouse lines were described elsewhere[9,12,30]. All mice were maintained in mixed background (C57BL/6× 129). The mice were housed in a 12 h light: dark cycle between 18 and 26 °C temperature with relative humidity of 30–70% and given ad libitum access to food and water for the duration of the study. Noon of the day of the vaginal plug was considered as E0.5. Mice and embryos were genotyped by polymerase chain reaction (PCR) using genomic DNA extracted from tail biopsy specimens and yolk sacs, respectively. Mice work at the Institut de Recherches Cliniques de Montréal (IRCM) was reviewed and approved by the IRCM animal care committee (protocols 2015-14 and 2017-10) in accordance with Canadian regulations. We have complied with all relevant ethical regulations.

**ChIP and sequencing**. HOXA11 ChIP was performed in forelimb buds of CD1 (wild type) and *Prx1Cre; Rosa^{Hoxa11/Hoxa11}* (*R26^{A11d/A11d}*) mice at E11.5 in the same conditions as described for HOX13 ChIP[11]. Chromatin was cross-linked using a combination of disuccinimidyl glutarate (DSG) and formaldehyde and

sonicated using Fisher Scientific, Model 100 sonic dismembrator to obtain fragments between 100 and 600 bp. Protein A and Protein G Dynabeads (Invitrogen) were incubated for 6 h at 4 °C with 5 µg HOXA11 (SAB1304728, Sigma) antibody. The chromatin was coupled to the beads overnight at 4 °C. The immunoprecipitated samples were then sequentially washed in low salt (1% Triton, 0,1% sodium dodecyl sulfate (SDS), 150 mM NaCl, 20 mM Tris (pH8), 2 mM EDTA), high salt (1% Triton, 0,1% SDS, 500 mM NaCl, 20 mM Tris pH8, 2 mM EDTA), LiCl (1% NP-40, 250 mM LiCl, 10 mM Tris (pH8), 1 mM EDTA) and TE buffer (50 mM NaCl, 10 mM Tris (pH8), 1 mM EDTA). The DNA was then purified on QIAquick columns (Qiagen). Library and flow cells were prepared by the IRCM Molecular Biology Core Facility according to Illumina's recommendations and sequenced on Illumina Hiseq 2500 in a 50 cycles paired-end configuration.

**ATAC-seq**. Dissection of proximal and distal forelimb buds from wild-type embryos and distal forelimb buds of *Hox13^{−/−}* embryos[11] were performed at E11.5. All samples for ATAC-seq were processed as described[21]. Briefly, 50,000 cells were washed in phosphate-buffered saline (PBS) and incubated on ice for 30 min in a hypotonic cell lysis buffer (0.1% w/v sodium citrate tribasic dehydrate and 0.1% v/v Triton X100) centrifugated (5 min at 2000*g* at 4 °C). Cells were then incubated 30 min on ice in cell lysis buffer (10 mM Tris-HCl, pH7.4, 10 mM NaCl, 3 mM MgCl₂, 0.1% v/v IGEPAL CA-630. After centrifugation (5 min at 2000*g* at 4 °C) the resulting pellet of nuclei was resuspended in transposase Master Mix (1.25 µl 10× TD buffer, 5 µl H₂O and 6.5 µl of Tn5: Illumina Nextera Kit; FC-121-1031) and incubated for 30 min at 37 °C. Samples were purified using MinElute PCR purification column (Qiagen). The eluted DNA was enriched and barcoded for multiplexing of samples using Nextera barcodes by PCR using Phusion kit. The library was recovered with GeneRead Purification columns. Samples were then evaluated by qPCR to test for the enrichment of open regions and sequenced on Illumina Hiseq 2500 with 50 bp paired-end reads, according to Illumina's recommendation.

**Single-cell ATAC-seq**. Dissection of forelimb buds from wild type and *Hox13^{−/−}* embryos[11] were performed at E11.5. Forelimbs were dissected in cold 1× PBS, after centrifugation at 300 rcf for 5 min at 4 °C the forelimbs were incubated in dissociation buffer (450 µl of 0.25% Trypsine/EDTA (GIBCO), 50 µl 10% bovine serum albumin (BSA), 1 µl DNAseI (NEB)) for 10 min at 37 °C. After incubation limb cells were gently mixed by pipetting up and down 10–15 times until they were dissociated, then 10% final fetal bovine serum was added. Dissociated cells were filtered using a cell strainer (40 µm Nylon, BD Falcon) and after centrifugation at 300 rcf for 5 min at 4 °C the resulting pellet of cells was resuspended in 1× PBS, 0.1% BSA. Dissociated cells were counted and assessed for cell viability using 0.4% Trypan blue. After centrifugation at 300 rcf for 5 min at 4 °C, the cell pellet was

resuspended in 100 µl of ice-cold lysis buffer (10 mM Tris-HCl, pH7.4, 10 mM NaCl, 3 mM MgCl2, 0.1% Tween-20, 0.1% NP-40, 0.01% Digitonin, 1% BSA) and incubated for 5 min on ice. Then 1 ml of wash buffer (10 mM Tris-HCl, pH7.4, 10 mM NaCl, 3 mM MgCl2, 0.1% Tween-20, 1% BSA) was added to the cells. After centrifugation at 300 rcf for 5 min at 4 °C, the pellet of nuclei was resuspended in diluted nuclei buffer in order to have 3500 nuclei/µl and processed using Chromium Single Cell ATAC Reagent Kits (10× Genomics, Pleasanton, CA) following the manufacturer recommendation. Briefly, nuclei were incubated in a Transposition Mix that includes a Transposase that preferentially fragments the DNA in open regions of the chromatin and simultaneously, adapter sequences were added to the ends of the DNA fragments. GEMs were then generated, using partitioning oil, in order to contain a single nucleus and a Master Mix to produce 10× barcoded single-stranded DNA after thermal cycling. The chromium single-cell ATAC Library was amplified and double sided size selected. Samples were controlled at multiple steps during the procedure by running on BioAnalyzer. Libraries were sequenced on Hiseq 4000 with 100 bp paired-end reads.

**RNA preparation and sequencing.** Dissection of proximal and distal of forelimb buds were performed at E11.5 as described above. The dissected limb buds were stored at −80 °C in Qiagen RNAlater until genotyping was performed. RNA was extracted from two independent embryos and performed in biological duplicate using RNAeasy Plus mini kit (Qiagen 74134). Ribosomal depletion, library preparation and flow-cell preparation for sequencing were performed by the IRCM Molecular Biology Core Facility according to Illumina's recommendations. Sequencing was done on a HiSeq 2500 instrument with a paired-end 50 cycles protocol.

**ChIP-seq and ATAC-seq data analysis.** ChIP-seq and ATAC-seq reads were aligned to the mm10 genome using bowtie v.2.3.1 with the following settings: bowtie2 -p 8 --fr --no-mixed --no-unal. Sam files were converted into tag directories using HOMER v4.9.1 (ref. [31]) and into bam files using Samtools v1.4.1 (ref. [32]) view function. Peaks were identified by comparing each sample to its input using MACS v2.1.1.20160309 (ref. [33]) callpeak function using the parameters: --bw 250 -g mm --mfold 10 30 -p 1e−5. For HOXA11 ChIP, peaks found in both replicates with a $p$ value < $10^{-20}$ were considered for further analysis. For HOX13 peaks, HOXA/D13 peaks underwent the same stringent filter. To obtain a high confidence list of specific peaks we considered binding as a specific peak when it did not overlap any peak with a $p$ value of $10^{-5}$ from either replicate of the compared ChIP. This strategy likely eliminates weakly specific or enriched binding (false negative) but allowed to focus on highly pure lists (true positive) of common vs. specific peaks.

Heatmaps and average profiles were generated using the Easeq software[34]. ChIP-seq and ATAC-seq data were visualized on the IGV software[35] using BigWig files generated using the makeUCSCfile HOMER command. For ATAC-seq differential analysis, peaks from ATAC datasets (proximal wild type and distal wild type for the proximo-distal comparison; distal wild type and distal $Hox13^{-/-}$ for the WT vs. $Hox13^{-/-}$ comparison) were merged using HOMER v4.9.1 mergePeaks tool to obtain a file with all the unique position from all the ATAC-seq datasets. ATAC-seq signals were quantified in these different datasets using the analyzeRepeats.pl HOMER command and differential accessibility analyses were performed using getDiffExpression.pl with default parameters, which uses DESeq2 (ref. [36]) to perform differential enrichment analysis. Peaks showing differential accessibility of more than two folds and an adjusted $p$ value smaller than 0.05 were considered differentially accessible. Association of ATAC-seq peaks with closest genes was analyzed using GREAT software[37].

The ChIP-seq data for HOX13 in distal limb at E11.5[11] were retrieved using Sratoolkit v2..8.2-1 from the NCBI Gene Expression Omnibus repository under the accession numbers GSE81356 and the Chicken ChIP-seq data of HOXA13 and HOXA11[7] from the accession numbers GSE86088.

**Motif analysis.** Motif analysis on ChIP data was performed using a fixed 200 bp window around the peak center. Motif analysis on ATAC-seq data was performed on the whole peak window as the aim is to identify any factors associated with chromatin accessibility and these may not be at the center of the ATAC-seq peak. In all cases, HOMER findMotifsGenome command was used to perform de novo analysis against background sequences generated by HOMER that matches GC content. When the motif rank is not indicated, only the top scoring motif based on its $p$ value is shown.

**RNA-seq data analysis.** Strand specific paired-end reads were aligned to the mm10 reference genome using TopHat2 v2.1.0 (ref. [38]) with the parameters --rg-library "L" --rg-platform "ILLUMINA" --rg-platform-unit "X" --rg-id "run#many" --no-novel-juncs --library-type fr-firststrand -p 12. The resulted Bam files were converted to tagDirectory using HOMER and BigWig were produced using the makeUCSCfile HOMER command.

RNA reads quantification was performed using HOMER analyzeRepeats.pl commands with the parameters: rna mm10 -count exons -condenseGenes -noadj -strand -.

Differential expression analysis was performed using HOMER getDiffExpression.pl command with the default parameters which performs differential expression with DESeq2.

**Single-cell ATAC-seq analysis.** Sequencing reads were aligned using Cellranger-atac version1.1.0 (from 10×Genomics®) and the resulted sorted bam files were structured into hdf5 snapfiles (single-nucleus accessibility profiles) using snaptools version 1.4.1. Quality control analysis was performed using R_Bioconductor v3.5.1_3.7 and SnapATAC 1.0.0. (ref. [39]). Cells with logUMI in [3.5–5] and fragment overlapping peaks above 25% were selected. They were cleaned by eliminating bins overlapping with ENCODE Blacklist regions[40], mitochondrial DNA as well as the top 5% of invariant features (house-keeping gene promoters). Dimensionality reduction was performed using a nonlinear diffusion map algorithm[41] available in the SnapATAC 1.0.0 package to produce 50 eigen-vectors of which the first 26 were selected (see Supplementary Fig. 6e) in order to generate K Nearst Neighbor graph with $K = 15$. The clustering was performed using Leiden Algorithm[42] available in the package leidenalg version 0.7.0 with a resolution of 0.4 and 2 UMAP (uniform Manifold Approximation and Projection version 0.2.3.1) embedding were generated using umap-learn v0.3.10 to visualize the data[43]. Fragments originating from the cells belonging to them same clusters were pooled using snaptools 1.4.1 and peak calling was performed using MACS2 version 2.1.2 for each of the clusters[44]. In order to create a reference peak list NarrowPeaks from each cluster of both WT and $Hox13$ mutant were merged using Bedtools 2.28.0, the resulting list was added to the hdf5 snapfiles using the snap-add-pmat function available in snaptools v1.4.1 in order to generate a cell by peak matrix which was subsequently loaded into the r snap-object. Differentially accessible regions (DARs) between the different clusters were identified using the findDAR function available in SnapATAC package v1.0.0 with a BCV of 0.1 (biological coefficient of variation) and which relies internally on the exactTest function from EdgeR v.3.14.0 to evaluate the significance of the difference in accessibility. The 2000 DARs with the lowest $p$ value and highest log fold change were further processed though Homer v4.11 to recover cluster-specific motifs (using a scanning window of 300 bp).

The HOXA11- and HOX13-specific binding sites coordinates obtained from our previous ChIPseq experiments were loaded on our hdf5 snapfiles (using Snaptools as described above) to generate a cell by peak matrix then added to the R snap-object; subsequently chromatin accessibility at specific binding sites was computed using a standard scoring method ($z$-score). Finaly, association of scATAC-seq peaks with closest genes was analyzed using GREAT software[37].

**Whole-mount in situ hybridization.** Whole-mount in situ hybridization was performed using standard procedure[45]. Generation of the $Bmp2$ probe was described elsewhere[46]. Briefly, embryos were rehydrated through a methanol series (100–30%), washed in PBST (0.1% Tween) and bleached for an hour on ice using 6% hydrogen peroxide before undergoing Proteinase K treatment for 15 min at RT and refixed with 4% formaldehyde. Next, embryos were hybridized with the RNA probes (5× SSC pH 4.5; 50% deionized formamide; 1% SDS; 0.1% Tween; 5 mg/mL torula RNA, 0.5 mg/mL heparin) overnight at 68 °C. Embryos were then washed with 1× TBS; 0.1% Tween, treated with 10% Goat serum; 1% BSA and incubated with alkaline phosphatase-conjugated anti-DIG antibodies (1/3000; Roche) overnight at 4 °C. The coloration was done using nitrotetrazolium blue chloride (NBT)/ 5-bromo-4-chloro-3-indolyl-phosphate, 4-toluidine salt (BCIP) substrate (Roche). After staining, the samples were washed in PBS and post-fixed. Embryos were photographed using the Leica, Wetzlar, Germany M165FC stereomicroscope coupled to the DFC450.C camera. Embryos were genotyped prior to experimentation and WT and $Hox13^{-/-}$ were treated identically in the same assay for comparison. A minimum of three embryos per genotype was assayed for reproducibility ($n = 3$).

**Protein extraction and western-blot analysis.** Nuclear extracts were performed using pooled forelimb and hindlimb buds at E11.5 from wild-type and $Hoxa11^{-/-}$ embryos. Western blot was performed using the anti-HOXA11 antibody (1:500) (SAB1304728, Sigma) and the anti-H3 antibody (1:3000) (abcam) was used as loading control (Supplementary Fig. 1).

**Reporting summary.** Further information on research design is available in the Nature Research Reporting Summary linked to this article.

## Data availability
The ChIP-seq data for HOX13 in distal limb and RNA-seq for wild type and $Hox13^{-/-}$ distal limb buds[11] at E11.5 are available from the NCBI Gene Expression Omnibus repository under the accession numbers GSE81356. Chicken ChIP-seq data of HOXA13 and HOXA11[7] were obtained from the accession numbers GSE86088. Raw files and processed files of all ChIP-seq, ATAC-seq and scATAC-seq can be found on GEO under the accession number GSE123482 [https://www.ncbi.nlm.nih.gov/geo/query/acc.cgi?acc=GSE123482] and GSE145657 [https://www.ncbi.nlm.nih.gov/geo/query/acc.cgi?acc=GSE145657]. The source data underlying Figs. 1c, f, 2a, c, d, f, g, and 3d; and Supplementary Figs. 2h, 3b, c, 4a, d–f, and 5b, c, e are provided as a Source Data file. Full scans of the blots are available in Supplementary Fig. 11. All other relevant data

supporting the key findings of this study are available within the article or from the corresponding author upon reasonable request.

## Code availability

All script used for the single-cell ATAC-seq analysis is available through Github (https://github.com/BCYasser/DigitMorphoScATAC/blob/master/analysis_script).

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

## Acknowledgements

We are particularly grateful to Guillaume Andrey for critical reading of the manuscript, lab members for insightful discussions and sharing reagents, Odile Neyret for NGS analyses and Jessica Barthe for mouse management. This work was supported by the Canadian Institute for Health Research (MOP-115127 and -126110 to M.K and FRN-154297 to J.D.) and ERC advanced grant INTEGRAL (ID 695032) to R.Z. Bioinformatics analyses were enabled in part by support provided by Calcul Quebec (www.calculquebec.ca), Compute Canada (www.computecanada.ca) and the Scientific IT and Application Support Center of EPFL. Y.K. was supported by a fellowship from the Molecular Biology program of the Université de Montréal and the IRCM fellowship Michel-Bélanger. A.M. was supported by an IRCM Challenge fellowship, C.G. was supported by the Jacques Gauthier IRCM fellowship, and I.D. is supported by the IRCM-Jean Coutu fellowship.

## Author contributions

Y.K., A.M., and M.K. conceived the study. I.D., Y.K., A.M. designed the experiments. I.D. and Y.K. with the help of C.G. and A.M. conducted the experiments. R.S. and R.Z. provided the data shown in Supplementary Fig. 10. A.M. with the help of I.D. conducted the bioinformatic analyses of all bulk datasets. Y.B. performed bioinformatics analyses of the single-cell ATAC-seq data set. I.D., Y.K., A.M., and M.K. analyzed and interpreted the data with the help of J.D. M.K. wrote the paper together with A.M., I.D., and Y.K. All authors commented on the paper.

## Competing interests

The authors declare no competing interests.

**Additional information**

