## [Peer Review File · Nature Communications]

Reviewers' comments:

Reviewer #1 (Remarks to the Author):

The manuscript by Desanlis et al. aims to describe a pioneer function of Hox13 transcription factors in the development of limb distal region that is required for the pentadactyl state. For this, they first analyze Hoxa11 and Hoxa13 chromatin binding in the developing limb and show that overexpression of Hoxa11 leads to Hoxa11 binding to Hoxa13-specific sites. Then, they explore a possible pioneer function of Hox13 TFs by performing ATAC-seq in Hox13^{-/-} distal late limbs, finding a set of regions with Hox13-dependent accessibility. Finally, they show that accessibility of the Hoxa11 enhancer in the distal limb, which is essential for the pentadactyl state, relies on Hox13 activity and is conserved in tetrapods.

For this work to be a strong candidate for publication in Nature Communications, several important issues have to be addressed, including a more solid demonstration of the Hox13 pioneer function during distal limb development.

a) General points:

1- The major problem of this work is that its main claim, the pioneer function of Hox13 TFs, is not clearly demonstrated. The decrease in chromatin accessibility in the absence of Hox13 does not necessarily imply a pioneer activity. Demonstration of the pioneer function of this TF also requires showing that Hox13 binds to non-accessible chromatin that becomes open later and fails to open in its absence. This could be assessed by performing ChIP-seq of Hoxa13 and ATAC-seq in early limb buds and comparing with the presented data of later limb development.

In addition, the authors need to show that the pioneer function has an impact over the distal limb genetic program by, using in their RNA-seq data, analyzing the GO and expressions changes of the genes associated to the differentially open regions.

2- Regarding the relationship between Hoxa11 and Hoxa13, it is assumed throughout the manuscript that binding of Hoxa11 to Hoxa13-specific targets when it is expressed in the distal limbs depends on Hoxa13 chromatin opening. However, there is no evidence of such dependency, since the presented ChIP-seq data of Hoxa11 ectopic expression in the distal limb is performed in the presence of Hox13. ChIP-seq of Hoxa11 in Hox13^{-/-} limbs has to be provided to demonstrate such dependency.

3- In general, the data presented could be analyzed much deeper to strongly demonstrate that Hox13 pioneer activity is essential for the distal limb program. Taking advantage of the RNA-seq data in proximal vs. distal limbs, as well as distal WT vs. distal Hox13^{-/-}, would help to strengthen the conclusions. In addition, more data have to be shown, including heatmaps and/or tracks, full tables with GOs, peaks, motifs, etc. Please, see the next points.

b) Specific points:

Fig. 1D: are the Hoxa11 and Hoxa13 specific peaks associated to differentially expressed genes between the proximal and distal limb buds?

Fig. 1F: a similar motif is found by de novo search in the 3 groups of peaks. Are these motifs similar to any known motif? In addition, which known motifs are enriched in those sets of peaks?

Fig. 1G: are 100% of the Hoxa13-specific sites also occupied by Hoxd13? If not, how many are co-occupied? Is this Hoxa13-Hoxd13 overlap different in Hoxa11-Hoxa13 common peaks? This is important, provided that the pioneer activity is attributed by the authors both to Hoxa13 and

Hoxd13.

Fig. 2B – Which genes and binding motifs are associated to the RA11KI-specific Hoxa11 sites? Are the associated genes more expressed in the distal vs. proximal limb?

Suppl Fig 2 – More than a half of the gained Hoxa11 sites in RA11KI limbs are occupied by Hoxa13, but not the rest. This raises the possibility that Hoxa11 binding in the distal limb may depend on additional factors distinct from Hoxa13. Again, the ChIP-seq of Hoxa11 in Hox13^{-/-} limbs would be required.

Fig. 3B: the differentially open regions have to be analyzed further. Which genes are associated with decreased accessibility? Are they enriched in distal limb development or differentially expressed in distal vs. proximal limbs or in Hox13^{-/-} vs WT distal limbs? In addition, which binding motifs are enriched? Is the Hox13 motif there? Its presence in the “decrease” group is essential to probe the pioneer activity.

Fig. 3B-E: heatmaps and example tracks showing the ATAC-seq signal of differentially open regions in Hox13^{-/-} vs. WT distal limbs, as well as Hoxa13 and Hoxa11 binding in those regions, have to be provided.

Line 134: authors say that “the distal gain of Hoxa11 expression in Hox13^{-/-} limbs results in increased accessibility within the T-DOM regulatory landscape”. This claim is too strong. First, whereas Hoxa11 expression extends to distal limb in Hox13^{-/-}, and there is increased accessibility of the T-DOM, it does not imply that Hoxa11 expression is responsible for this increased accessibility.

Fig. 4D – The ChIP-seq and ATAC-seq data from chick limb buds should be further used. Authors use them to demonstrate evolutionary conservation of the chromatin accessibility and Hox13 dynamics in the Hoxa11 enhancer. However, these data should be deeper exploited and compared with mouse data throughout the manuscript to strengthen the conclusion that the function of Hox13 in the distal limb is conserved in tetrapods.

c) Minor points:

- Color scales for heatmaps should be provided.
- Fig. 1D: please, show the complete names for the GO terms.
- Fig. 4A: the polarity of the strand-specific RNA-seq signal (+ or -) seems to be swapped.

Reviewer #2 (Remarks to the Author):

This study examines the chromatin-binding activities of Hox13 proteins, key regulatory transcription factors (TFs) in distal limb morphogenesis. The data are convincing, and I have no experimental criticisms of the paper. Moreover, the results provide critical insight into the mechanisms by which these key TFs orchestrate downstream networks of genes in establishing limb pattern. Nonetheless, I have mixed feelings about this contribution.

On the one hand, the central finding is important. We know that the same TFs can trigger very different transcriptional and functional responses in the context of different cell types. While partially attributable to the presence or absence of various co-factors and binding partners, a major determinant of the impact of a TF relates to which regions of the chromatin are accessible to it (in an “open” state) in a given cell type. Importantly, then, as cells differentiate, previously

inaccessible regions of heterochromatin must be opened to allow new genes to be activated by the TFs present in the cell. This is accomplished, in part, through the activity of so-called "pioneer" TFs, that can bind to both open or closed chromatin. Once bound to the DNA, pioneer TFs can recruit other factors, thereby nucleating a state-change in a specific region of the chromosome, ie making genes in the region around their binding site accessible to transcriptional machinery. This short report demonstrates that, during limb development, the important patterning transcription factors Hoxa13 and Hoxd13 can function as pioneer factors. This is an important conceptual advance in our understanding. It is certainly appropriate for publication in Nature Communication.

On the other hand, the authors chose to frame their presentation around the observation that among the many loci in closed chromatin where the Hox13 genes are able to bind, is a putative enhancer previously shown to play a role in regulating the boundaries of Hox gene expression in the autopod and zeugopod. When this element is mutated, earlier studies showed that the resultant mice exhibit a variable polydactyly. The authors, inexplicably in my mind, make this the central focus of their current presentation. Even if the earlier conclusions about polydactyly correct, Hox genes do far more in distal limb patterning than limit digit number to five. Moreover, in their previous study of this enhancer element, the authors made the intriguing, yet fairly speculative suggestion that the enhancer in question played a role in the evolution of a stable pentadactyl pattern in modern tetrapods. Putting this together with their current study, the authors now make an unwarranted extrapolation, to suggest that the evolution of the pioneer function of Hox13 genes was central to the fin-to-limb transition when pentadactyly became canalized. I certainly do not object to the authors making such speculations (if resented as such) within the Discussion. However, it does not belong in the Title, the Abstract and/or the Introduction as well. Not only is it extremely misleading for a reader outside the field, it also obscures other, important implications of Hox13 genes being pioneer factors.

To put it differently, if the main conclusion the authors want to draw is that pioneer activity of Hox13 genes is key to pentadactyly (the title), then the data fall short of supporting their claim and the paper should not be accepted. However, if the main conclusion is that Hox13 genes, play a central role in the transition from zeugopod to autopod by virtue of their ability to act as pioneer factors, then I am very supportive of publication, following changes limited to the text.

Minor point: Under author contribution, the roles of three of the authors are not listed (Sheth, Zeller and Drouin). These authors may have simply contributed mouse lines or other reagents, but their contribution needs to be clarified.

We are thankful for the comments and suggestions made by the reviewers to improve the quality of our manuscript. As you will see, we have made significant changes to the manuscript to address the concerns raised. Notably, we have substantially revised the title, abstract and main text to focus on the role of the HOX13-dependent chromatin accessibility in the transition from the proximal to the distal limb development program as suggested by reviewer 2, we have extended the analysis of the sequencing data provided in the previous version of the manuscript and added the study of HOXA11 binding in the HOX13 null limb bud as requested by reviewer 1. We also added single cell ATAC-seq data and analysis of proximal versus distal limb chromatin accessibility to substantiate that HOX13-mediated chromatin accessibility at HOX13 targets establishes the distal limb-specific chromatin accessibility landscape.

Point-by-point responses to reviewer comments are below. (*Reviewers' comments literatim are in italics*)

Reviewer #1 (Remarks to the Author):

The manuscript by Desanlis et al. aims to describe a pioneer function of Hox13 transcription factors in the development of limb distal region that is required for the pentadactyl state. For this, they first analyze Hoxa11 and Hoxa13 chromatin binding in the developing limb and show that overexpression of Hoxa11 leads to Hoxa11 binding to Hoxa13-specific sites. Then, they explore a possible pioneer function of Hox13 TFs by performing ATAC-seq in Hox13-/- distal late limbs, finding a set of regions with Hox13-dependent accessibility. Finally, they show that accessibility of the Hoxa11 enhancer in the distal limb, which is essential for the pentadactyl state, relies on Hox13 activity and is conserved in tetrapods.

For this work to be a strong candidate for publication in Nature Communications, several important issues have to be addressed, including a more solid demonstration of the Hox13 pioneer function during distal limb development.

a) General points:

1- The major problem of this work is that its main claim, the pioneer function of Hox13 TFs, is not clearly demonstrated. The decrease in chromatin accessibility in the absence of Hox13 does not necessarily imply a pioneer activity. Demonstration of the pioneer function of this TF also requires showing that Hox13 binds to non-accessible chromatin that becomes open later and fails to open in its absence. This could be assessed by performing ChIP-seq of Hoxa13 and ATAC-seq in early limb buds and comparing with the presented data of later limb development.

We thank the reviewer for highlighting that the demonstration of HOX13 pioneer activity could have been more strongly substantiated.

In the revised version of the manuscript, we added a number of novel data to address this point.

1. We now compare ATAC-seq data from proximal and distal limb bud and show that loci with distal-specific chromatin accessibility are bound by HOX13 (new Figure 2a-c).
2. Importantly, these HOX13 targets, whose chromatin accessibility is specific to distal limb bud, lose their accessibility upon HOX13 inactivation (new Figure 2f).
3. We also substantiate this genome wide action of HOX13 by showing specific examples of loci relevant to distal limb development (Fig. 2h-i, Suppl. Fig. 5f).
4. In addition, and in relation to point 2 below, we demonstrate that HOX13-dependent chromatin accessibility allows the binding of (at least one) other transcription factor (Fig. 1f, Fig. 2g-i, Suppl. Fig. 3f), which is a key characteristic of pioneer factors.

As HOX13 expression is restricted to the late distal limb, demonstrating that HOX13 TFs bind at inaccessible loci before these loci switch to an accessible state requires performing ChIP-seq analysis at stages of initial HOX13 expression in the distal limb bud. However, this is technically challenging because the initial expression of HOX13 occurs in a small cell population at the distal tip of the limb bud to gradually expand to the entire distal limb as development proceeds. In absence of morphological features characterizing the initial HOX13 expression, there is no means to isolate cells at the same stage of HOX13 expression and in sufficient amount to perform ChIP-seq experiments.

However, data from two recent manuscripts posted on BioRxiv provide evidence that HOX13 TFs are able to bind inaccessible loci before these loci switch to an open conformation:

1. By studying the HoxD regulatory landscape associated with genital bud development, a tissue in which HOX13 activation is homogenous, Amandio et al. (bioRxiv 810788, 2019) identified HOX13 binding at an

enhancer (referred to as Prox) in an inaccessible state at the stage of initial HOX13 expression, and this enhancer switches to an accessible state at subsequent stages of development.

2. Using the in vitro motor neurons differentiation system to study HOXC binding specificity, Bulajic et al. (bioRxiv 2019.12.29.890335) provide evidence that HOXC13 binds loci in a closed chromatin conformation at early stages following transfection and these loci become accessible at later stages.

These evidence that HOX13 can bind to loci before these loci switch to an accessible state together with our in vivo data revealing that HOX13-dependent chromatin accessibility at HOX13 targets establishes the distal limb-specific chromatin accessibility landscape, strongly support HOX13 TFs being pioneer factors. We now discuss these two new papers in our manuscript to further substantiate our conclusion that HOX13 TFs have functional properties characteristic of pioneer factors.

In addition, the authors need to show that the pioneer function has an impact over the distal limb genetic program by, using in their RNA-seq data, analyzing the GO and expressions changes of the genes associated to the differentially open regions.

As suggested by the reviewer, we have now performed additional analyses which show that:

1. The most enriched GO terms associated with the HOX13-dependent chromatin accessibility includes skeletal system development, limb development and digit morphogenesis (new Suppl. Figure 5d)
2. Distal limb specific chromatin accessibility are sites bound by HOX13 and 2/3 of the distal limb genes are downregulated in *Hox13*^{-/-} mutant (new Suppl. Figure 5g).
3. Genes expressed in distal limbs are located closer to distal-enriched accessible sites than proximal enriched accessible sites (new Suppl. Figure 4f)

It should be noted that the vast majority of loci bound by HOX13 are distal regulatory elements, i.e. not promoters (Sheth et al. 2016 and new Suppl. Figure 2h) and because regulatory elements can act over large distance, target genes for most of these regulatory elements remain to be established. Nonetheless, we provide several examples of HOX13-dependent chromatin accessibility at enhancers which were previously shown to control genes associated with distal limb development (new Fig. 2h-i; Fig. 3d-e, Suppl. 5f), which are down-regulated in the *Hox13* mutant limb.

Importantly, we now demonstrate the relevance of HOX13-dependent chromatin accessibility to the distal limb-specific chromatin accessibility landscape using single cell ATAC-seq (new Figure 3). Comparison between wild type and *Hox13*^{-/-} limb buds shows that cells characterized by the distal limb-specific chromatin accessibility profile in the wild type context are the cells with a modified chromatin accessibility landscape in the *Hox13* null limb bud.

*2- Regarding the relationship between *Hoxa11* and *Hoxa13*, it is assumed throughout the manuscript that binding of *Hoxa11* to *Hoxa13*-specific targets when it is expressed in the distal limbs depends on *Hoxa13* chromatin opening. However, there is no evidence of such dependency, since the presented ChIP-seq data of *Hoxa11* ectopic expression in the distal limb is performed in the presence of *Hox13*. ChIP-seq of *Hoxa11* in *Hox13*^{-/-} limbs has to be provided to demonstrate such dependency.*

We thank the reviewer for this comment, and the dependency of HOXA11 was indeed assumed to be dependent on HOX13. We have now performed HOXA11 ChIP-seq in *Hox13*^{-/-} limb buds (i.e. *Hoxa13*^{-/-};*Hoxd13*^{-/-} limb buds) demonstrating that the ectopic binding of HOXA11 at HOXA13-specific sites is lost upon HOX13 inactivation, thereby showing that this ectopic binding occur only when these loci are in an accessible chromatin conformation (Figure 1f, Fig. 2g) while the wild type HOXA11 binding is unaffected (Suppl. Fig.3b-c).

We wish to highlight that crosses to obtain these double homozygous mutant embryos can only be made with double heterozygous mice and these mice are hypofertile. It has thus been a major commitment to produce enough *Hox13* mutant limb buds to perform the HOXA11 ChIP-seq.

*3- In general, the data presented could be analyzed much deeper to strongly demonstrate that *Hox13* pioneer activity is essential for the distal limb program. Taking advantage of the RNA-seq data in proximal vs. distal limbs, as well as distal WT vs. distal *Hox13*^{-/-}, would help to strengthen the conclusions. In addition, more data have to be shown, including heatmaps and/or tracks, full tables with GOs, peaks, motifs, etc. Please, see the next points.*

b) Specific points:

*Fig. 1D: are the *Hoxa11* and *Hoxa13* specific peaks associated to differentially expressed genes between the proximal and distal limb buds?*

As shown in the new Suppl. Fig.2h, HOXA11 and HOXA13 bind intergenic regulatory regions. Because these regulatory elements can work over large distance, in most cases, their *bona fide* target gene(s) remain unknown. Yet,

for the regulatory elements for which the targets have been experimentally validated, our data show that HOXA13-specific binding is associated with distally expressed genes (see new Fig. 2h-i, 3d). As for the HOXA11 specific peaks, we show the example of the experimentally validated Alx4 proximal enhancer (Kuijper et al., Developmental Biology 2005) as well as candidate enhancers of genes expressed in the proximal limb (new suppl. Fig. 2i).

Fig. 1F: a similar motif is found by de novo search in the 3 groups of peaks. Are these motifs similar to any known motif? In addition, which known motifs are enriched in those sets of peaks?

The motifs identified by de novo search in the 3 groups are similar to the Hox type motif with some minor variations between the two sets of specific peaks. For the HOXA11-specific peaks, the other known motif is a T-box motif, while a Lhx motif is the other known motif found both at the HOXA13-specific and the common peaks. This is now indicated in Figure 1d.

Fig. 1G: are 100% of the Hoxa13-specific sites also occupied by Hoxd13? If not, how many are co-occupied? Is this Hoxa13-Hoxd13 overlap different in Hoxa11-Hoxa13 common peaks? This is important, provided that the pioneer activity is attributed by the authors both to Hoxa13 and Hoxd13.

In the new Suppl. Figure 2, we show the strong stringency that was used for comparing ChIP-seq data, as a way to avoid contaminating our lists with false specific peaks. To do so, we have only used peaks with a pvalue < 10⁻²⁰ and considered peaks specific when they were absent from the general list of peaks (pvalue < 10⁻⁵). In the case of HOXA13 and HOXD13, >95% of HOXA13 peaks are occupied by HOXD13 and vice versa. We have also clarified this point by adding HOXD13 ChIP-seq data in every figure where HOXA13 ChIP-seq is shown (new Figures 1c, 1f, 2f, 2h, 2i, 3d, Suppl. Figures 2h, 3a, 3b, 3d, 3e, 5b, 5g).

Fig. 2B – Which genes and binding motifs are associated to the RA11KI-specific Hoxa11 sites? Are the associated genes more expressed in the distal vs. proximal limb?

Suppl Fig 2 – More than a half of the gained Hoxa11 sites in RA11KI limbs are occupied by Hoxa13, but not the rest. This raises the possibility that Hoxa11 binding in the distal limb may depend on additional factors distinct from Hoxa13. Again, the ChIP-seq of Hoxa11 in Hox13-/- limbs would be required.

In RA11KI limbs (now renamed R26^{A11d/A11d} to be more consistent with the fact that the allele is the knock-in of Hoxa11 cDNA at the rosa26 locus), the conditional HOXA11 expression is driven by the Prrx1-cre allele, which results in ectopic HOXA11 expression in the distal limb but also in the very proximal limb domain (Kherdjemil et al., Nature 2016). The HOXA11 ChIP-seq in the R26^{A11d/A11d} mutant was made using the entire limb bud and thereby R26^{A11d/A11d}-specific HOXA11 sites include sites specific for the very proximal limb (presumptive stylopod, where HOXA11 is normally not expressed) in addition to the distal specific sites. It is thus not surprising that HOXA11 also binds ectopic sites that are distinct from HOX13 sites. Characterizing the R26^{A11d/A11d}-specific HOXA11 sites in the very proximal limb bud goes beyond the scope of this manuscript which focuses on HOX13-dependent chromatin accessibility. However, we have done the other study suggested by this reviewer, which was to perform ChIP-seq for HOXA11 in HOX13^{-/-} limbs. The results clearly show that HOXA11 ectopic binding at sites normally HOX13-specific and whose chromatin accessibility relies on HOX13, is lost upon HOX13 inactivation (New Figures 1f and 2g). This provides evidence that HOXA11 ectopic binding in distal limb requires HOX13-dependent chromatin accessibility.

Fig. 3B: the differentially open regions have to be analyzed further. Which genes are associated with decreased accessibility? Are they enriched in distal limb development or differentially expressed in distal vs. proximal limbs or in Hox13-/- vs WT distal limbs?

We now provide data clarifying that the differentially open regions in Hox13^{-/-} limbs (with decreased accessibility) are regions with distal-specific chromatin accessibility and are bound by HOX13 TFs (new Figure 2d-f). As HOX13 TFs are *bona fide* markers of distal limb development (Fromental-Ramain et al. Dev 1996, Scotti et al., Genesis 2015, Sheth et al. Cell Reports 2016), the fact that the differentially open regions are bound by HOX13 supports that these regions are associated with distal limb identity. This is further substantiated with the evidence that down-regulated genes in Hox13^{-/-} limbs are primarily distal specific genes (Suppl. Figure 5e, g).

We also provide examples of differentially open regions for which target genes are known (new Figure 2h-i; Figure 3d) and these genes are down-regulated in Hox13^{-/-} limbs (Suppl. Figure 5g).

In addition, GREAT analysis for the differentially open regions in Hox13^{-/-} limbs has now been performed and the most enriched GO terms include ‘embryonic digit morphogenesis’ (suppl. Figure 5d).

In addition, which binding motifs are enriched? Is the Hox13 motif there? Its presence in the “decrease” group is essential to probe the pioneer activity.

We now show the binding motifs enriched for the decrease accessibility group in the new Figure 2e. The most enriched motif (p value 1e-384) is a HOX motif almost identical to the HOX13-specific motif reported in Figure 1d.

Fig. 3B-E: heatmaps and example tracks showing the ATAC-seq signal of differentially open regions in Hox13-/- vs. WT distal limbs, as well as Hoxa13 and Hoxa11 binding in those regions, have to be provided.

We have now added several examples of gene tracks showing HOX13 pioneering action at functionally relevant loci and ChIP-seq tracks for HOXA11 and HOX13 (new Fig. 2h-i, new Suppl. Fig 5f, new Fig. 3d). We have also added heatmaps (new Figure 2f) and boxplots (new Figure 2g).

Line 134: authors say that “the distal gain of Hoxa11 expression in Hox13^{-/-} limbs results in increased accessibility within the T-DOM regulatory landscape”. This claim is too strong. First, whereas Hoxa11 expression extends to distal limb in Hox13^{-/-}, and there is increased accessibility of the T-DOM, it does not imply that Hoxa11 expression is responsible for this increased accessibility.

The reviewer is correct and we have removed this sentence.

Fig. 4D – The ChIP-seq and ATAC-seq data from chick limb buds should be further used. Authors use them to demonstrate evolutionary conservation of the chromatin accessibility and Hox13 dynamics in the Hoxa11 enhancer. However, these data should be deeper exploited and compared with mouse data throughout the manuscript to strengthen the conclusion that the function of Hox13 in the distal limb is conserved in tetrapods.

As suggested by reviewer number 2, we have modified the presentation of our data to focus on the role of HOX13-dependent accessibility in the implementation of the distal limb developmental program. Evolutionary considerations are now restricted to a brief paragraph at the end of the discussion and therefore the former Figure 4D is now part of the supplementary figures (Suppl. Fig.10).

c) Minor points:

- Color scales for heatmaps should be provided.

Color scales for heatmaps are now provided.

- Fig. 1D: please, show the complete names for the GO terms.

Complete names for the GO terms are now provided (see new Fig. 3; Suppl. Fig. 4c, 5d, 9)

- Fig. 4A: the polarity of the strand-specific RNA-seq signal (+ or -) seems to be swapped.

We thank the reviewer for highlighting this error. This has now been corrected (note that former Figure 4A is now the new Figure 2i).

Reviewer #2 (Remarks to the Author):

This study examines the chromatin-binding activities of Hox13 proteins, key regulatory transcription factors (TFs) in distal limb morphogenesis. The data are convincing, and I have no experimental criticisms of the paper. Moreover, the results provide critical insight into the mechanisms by which these key TFs orchestrate downstream networks of genes in establishing limb pattern. Nonetheless, I have mixed feelings about this contribution.

On the one hand, the central finding is important. We know that the same TFs can trigger very different transcriptional and functional responses in the context of different cell types. While partially attributable to the presence or absence of various co-factors and binding partners, a major determinant of the impact of a TF relates to which regions of the chromatin are accessible to it (in an “open” state) in a given cell type. Importantly, then, as cells differentiate, previously inaccessible regions of heterochromatin must be opened to allow new genes to be activated by the TFs present in the cell. This is accomplished, in part, through the activity of so-called “pioneer” TFs, that can bind to both open or closed chromatin. Once bound to the DNA, pioneer TFs can recruit other factors, thereby nucleating a state-change in a specific region of the chromosome, ie making genes in the region around their binding site accessible to transcriptional machinery. This short report demonstrates that, during limb development, the important patterning transcription factors Hoxa13 and Hoxd13 can function as pioneer factors. This is an important conceptual advance in our understanding. It is certainly appropriate for publication in Nature Communication.

On the other hand, the authors chose to frame their presentation around the observation that among the many loci in closed chromatin where the Hox13 genes are able to bind, is a putative enhancer previously shown to play a role in regulating the boundaries of Hox gene expression in the autopod and zeugopod. When this element is mutated, earlier studies showed that the resultant mice exhibit a variable polydactyly. The authors, inexplicably in my mind, make this the central focus of their current presentation. Even if the earlier conclusions about polydactyly correct, Hox genes do far more in distal limb patterning than limit digit number to five. Moreover, in their previous study of this enhancer element, the authors made the intriguing, yet fairly speculative suggestion that the enhancer in question played a role in the evolution of a stable pentadactyl pattern in modern tetrapods. Putting this together with their current study, the authors now make an unwarranted extrapolation, to suggest that the evolution of the pioneer

function of Hox13 genes was central to the fin-to-limb transition when pentadactyly became canalized. I certainly do not object to the authors making such speculations (if resented as such) within the Discussion. However, it does not belong in the Title, the Abstract and/or the Introduction as well. Not only is it extremely misleading for a reader outside the field, it also obscures other, important implications of Hox13 genes being pioneer factors.

To put it differently, if the main conclusion the authors want to draw is that pioneer activity of Hox13 genes is key to pentadactyly (the title), then the data fall short of supporting their claim and the paper should not be accepted. However, if the main conclusion is that Hox13 genes, play a central role in the transition from zeugopod to autopod by virtue of their ability to act as pioneer factors, then I am very supportive of publication, following changes limited to the text.

We thank the reviewer for this comment, which made us realize that the previous version of the text/title was misleading for readers outside the field. We have made substantial changes to the main text, abstract, figures and title and the revised version of the manuscript now focus on the role of the HOX13-dependent chromatin accessibility in the transition from the zeugopod to the autopod developmental program. We only briefly discuss the potential implication of HOX13 pioneer activity during evolution at the end of the discussion.

Minor point: Under author contribution, the roles of three of the authors are not listed (Sheth, Zeller and Drouin). These authors may have simply contributed mouse lines or other reagents, but their contribution needs to be clarified. The contribution of Drs Sheth, Zeller and Drouin has now been clarified in the author contribution section.

REVIEWERS' COMMENTS:

Reviewer #1 (Remarks to the Author):

The authors have done an excellent work addressing all my concerns and have provided very interested new data. I now strongly recommend the publication of this work in Nature Communication.

Reviewer #2 (Remarks to the Author):

The authors have done a nice job addressing the concerns of both referees. I am happy to support publication of this contribution in its current form.